# 🔏 TaskCraft: Automated Generation of Agentic Tasks

**Dingfeng Shi**[1]*, **Jingyi Cao**[1]*, **Qianben Chen**[1]*, **Weichen Sun**[1], **Weizhen Li**[1], **Hongxuan Lu**[1],
**Fangchen Dong**[1], **Tianrui Qin**[1], **King Zhu**[1], **Minghao Liu**[2,3], **Yuchen Eleanor Jiang**[1],
**Jian Yang**[2,4], **Ge Zhang**[2,5], **Jiaheng Liu**[2,6], **Changwang Zhang**[1], **Jun Wang**[1], **Wangchunshu Zhou**[1]†

[1]OPPO  [2]M-A-P  [3]2077AI  [4]Beihang University  [5]ByteDance  [6]Nanjing University
{shidingfeng,zhouwangchunshu}@oppo.com

## Abstract

Agentic tasks, which require multistep problem solving with tool use and adaptive reasoning, are becoming increasingly central to the advancement of NLP and AI. Although benchmarks such as GAIA and BrowseComp have advanced agent evaluation, their scalability remains limited by the high cost of human annotation. We introduce TaskCraft, the first automated workflow for generating scalable, multitool, and verifiable agentic tasks of difficulty. TaskCraft progressively complexifies atomic tasks through depth-based and width-based extensions, with incremental validation via rejection sampling and LLM-based linguistic analysis, ensuring both scalability and efficiency. The generated tasks enable trajectory sampling within state-of-the-art workflows, supporting end-to-end SFT and RL training. Experimental results on multiple LLMs show that TaskCraft data substantially improves multi-hop reasoning and agentic capabilities. Further scaling with TaskCraft tasks and applying RL training yields additional gains, achieving state-of-the-art performance on four agentic benchmarks. The resulting dataset comprises 41k tool-intensive tasks across varied difficulty levels, including 12.6k tool-interaction trajectories and 5k multihop decompositions. https://github.com/OPPO-PersonalAI/TaskCraft

## 1 Introduction

Agentic tasks, defined as autonomous multi-step problem solving that requires tool use and adaptive reasoning, are becoming increasingly central to AI and NLP. Recent progress in language agents Significant-Gravitas (2023); Wu et al. (2023); Li et al. (2023); Zhou et al. (2023a;b; 2024) empowered agentic workflows to address increasingly complex tasks. For example, ReAct Yao et al. (2023) adopts the Thought–Action–Observation (TAO) paradigm, enabling workflows to solve problems through iterative reasoning and repeated interaction with the environment.

To assess advanced agent capabilities, benchmarks such as GAIA Mialon et al. (2023), BrowseComp Wei et al. (2025), and Humanity's Last Exam (HLE) Phan et al. (2025) have been introduced. GAIA evaluates reasoning, tool use, and web browsing through 466 real-world questions. BrowseComp comprises 1,266 tasks that test an agent's ability to retrieve and integrate complex online information. HLE includes 2,500 multimodal questions across more than 100 disciplines to measure advanced reasoning and domain knowledge. Although these datasets have advanced agent evaluation, their scalability is constrained by the high cost of manual annotation. For instance, constructing HLE required 1,000 experts to label only 2,500 examples, making large-scale expansion impractical.

Previous work has explored the use of large language models to automatically generate queries, addressing the scarcity and annotation cost of human-labeled data. These queries can then support reasoning trajectory sampling for supervised fine-tuning (SFT) and Reinforcement Learning (RL). A representative example is the Self-Instruct framework Wang et al. (2022), which demonstrated that

---

*Equal contribution
†Corresponding author

LLMs can generate high-quality, diverse instruction data for multiturn dialogues. However, these methods are primarily designed for static instruction-following scenarios and fall short in modeling agentic tasks that require interaction with external tools and environments. Consequently, such data are insufficient for training or evaluating agents that operate in dynamic, real-world settings.

In this work, we introduce TASKCRAFT, the first agentic workflow for the automated generation of agentic tasks (queries), with a particular focus on tasks that require chain-of-tool execution. Our approach provides the following advantages:

- **Scalability.** The workflow supports adaptive difficulty, multi-tool integration, and the generation of tasks beyond the capabilities of the task-generation agent, along with their corresponding trajectories.

- **Efficient Verification.** During each task extension, only incremental components undergo agentic validation, eliminating the need for full verification of the extended task.

Our approach begins by generating atomic tasks solvable with single-tool invocations, then progressively increases their complexity through depth-based and width-based extensions. To ensure task quality, we apply rejection sampling to retain cases where agents with external tools succeed but LLMs fail, validating genuine tool necessity. LLM-based linguistic analysis accelerates validation by rapidly examining the incremental modifications introduced during task complexification, without requiring full execution of the entire task.

Based on this method, we constructed a dataset of about 41k candidate tasks spanning different difficulty levels. It further contains roughly 12.6k tool-interaction trajectories and around 5k instances of multi-hop sub-task decomposition.

To evaluate the effectiveness of our generated tasks, we build on the training data used in Tool-Integrated Reasoning (TIR) models Schick et al. (2023); Shen et al. (2023); Wu et al. (2025a) and augment it with TaskCraft-generated tasks for SFT and RL trajectory sampling. Incorporating these tasks consistently improves TIR model performance across multiple benchmarks. On GAIA, MHQA data yields 38.8%, which rises to 60.2% (+21.4) with 2.5k TaskCraft tasks and further to 60.8% (+22.0) with 8k TaskCraft tasks for RL, achieving state-of-the-art (SOTA) results among TIR models and demonstrating the effectiveness of our approach.

## 2 NOTATIONS AND PRELIMINARY

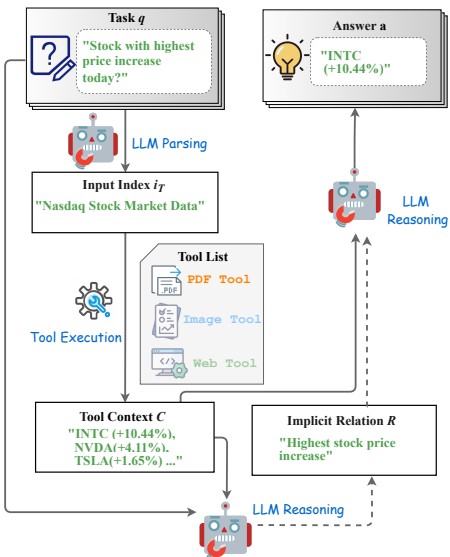

Figure 1: Execution flow of a single tool invocation.

To design tasks for agentic reasoning, we abstract the execution process of an agentic task. As shown in figure 1, given a task $q$, tool execution involves two stages: locating the input index $i_T$ (e.g., a stock data website) and operating the tool $T$ on it (e.g., a browser accessing the website). Executing $T$ with $i_T$ yields the context $C$ (e.g., stock price data), from which the LLM applies the relation $R$ specified in the task (e.g., identifying the highest stock price) to derive the final answer $a$.

An agentic task can thus be minimally defined by an input index $i_T$ and a relation $R$ over the tool-execution context. Since $R$ depends on the retrieved context $C$, the tool must be executed before the answer can be derived.

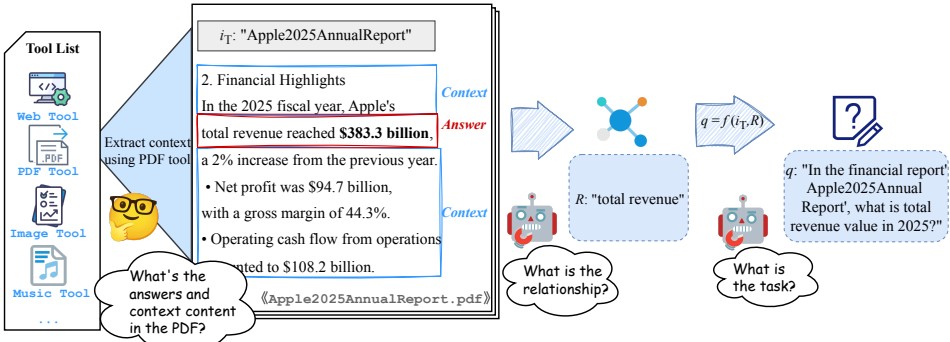

Figure 2: Atomic task generation. From an unlabeled corpus, we extract $i_T$ and derive textual content $C$ via tool execution. LLM identifies candidate answers $a$ from $C$, infers their relationship $R$, and constructs question $q$ conditioned on $i_T$ and $R$.

> **Atomic Task**
>
> An atomic task is resolved with a single target tool invocation. To simplify, we disregard search and file system operations, assuming a detailed input index $i_T$ enables retrieval through finite navigation.

Given an answer $a$, the most direct approach to construct an atomic task involves prompting an LLM to generate the corresponding question. However, questions produced in this manner often suffer from low tool invocation rates, unpredictable difficulty levels, unregulated tool requirements, and inconsistent verification complexity (see section 4.4 for more details).

To address these issues, we assume an ideal search engine that retrieves precise data based on $i_T$ (e.g., paper titles, song titles). Under this assumption, we define a task as $(q, a) = (f_q(i_T, R), a)$, where $f_q$ is a sampling function guiding the LLM to generate $q$ in natural language form by using $i_T$ and $R$.

## 3 Automated Task Generation Workflow

In this section, we describe our task construction workflow, which proceeds in three stages: (1) generating atomic tasks as the foundation, (2) progressively extending them to increase complexity, and (3) verifying their validity through efficient checks.

### 3.1 Atomic Task Generation

As figure 2 shown, we begin by compiling a corpus of unlabeled data aligned with the tool's input requirements. From this corpus, we extract $i_T$ and derive textual content $C$ via tool execution. For example, browsing, PDF, and image comprehension tools yield webpage titles, PDF names, and image paths, from which we extract textual content $C$ for answer sampling. We prompt an LLM to identify key candidate answers $a$ from $C$ and infer their relationship $R$ with $C$, ultimately constructing question $q$ conditioned on $i_T$ and $R$.

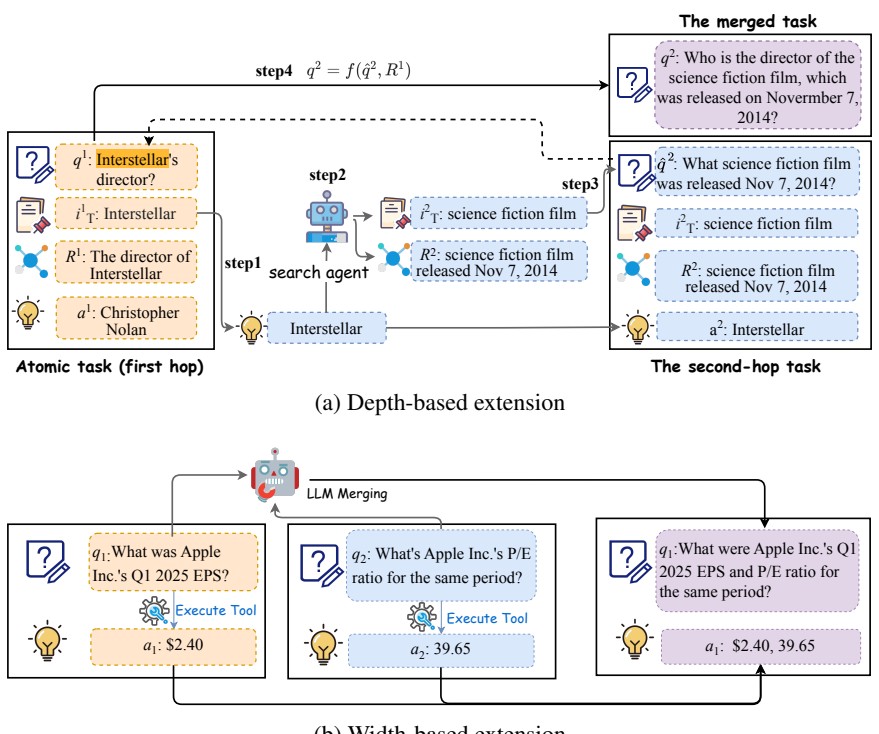

Figure 3: Strategy for task extension

## 3.2 TASK EXTENSION

In order to increase task difficulty in a scalable way, we adopted two extended task strategies: the *depth-based extension* and the *width-based extension*. (see Appendix E for prompts details)

**Depth-based extension.** We aim to construct tasks requiring multiple sequential tool executions, where each step depends on the output of the previous one. To achieve this, a new sub-task must be derived from a known task $q^n$. The tool input index $i_T$ at each stage exhibits strong extensibility due to (1) its frequent association with proper nouns, which are less likely to be memorized by LLMs, and (2) its natural suitability for recursive definition. Specifically, a n-hop task $(q^n, a)$, consisting of a question $q^n$ and its corresponding answer $a$, is formulated as follows:

$$(q^n, a) = (f_q(i_T^n, R^n), a), \tag{1}$$

To extend a n-hop task $q^n$ into a (n+1)-hop task $q^{n+1}$, we first find a intermediate sub-task:

$$(\hat{q}^{n+1}, i_T^n) = (f_q(i_T^{n+1}, R^{n+1}), i_T^n). \tag{2}$$

Here, $i_T^{n+1}$ (e.g., a song title) represents a new index derived from $i_T^n$ (e.g., a fragment of the song's lyrics) through reversible operations. To achieve this, a search agent identifies the *title of superset* of $i_T^n$ on the web or within the file system and uses it as $i_T^{n+1}$. We instruct the agent to search for a superset to reduce the risk of cyclic generation.

During the search, the agent retrieves the *supersets* text content $C$ (e.g., the complete lyrics of the song) with search tools. An LLM then analyzes $C$ to infer its relationship $R^{n+1}$ to $i_T^n$ (e.g., that the fragment corresponds to the third line of the lyrics).

Using this intermediate task, we can define the recursive formulation to obtain the (n+1)-hop task:

$$(q^{n+1}, a) = (f_m(q^n, \hat{q}^{n+1}, i_T^n), a), \tag{3}$$

where $f_m$ is a function that guides the LLM to generate $q^{n+1}$ in natural language by substituting $i_T^n$ in $q^n$ with $\hat{q}^{n+1}$.

**Width-based extension.** The goal of the width-based extension is to generate a new task that needs to be decoupled into multiple sub-tasks to be completed. For simplicity, for two sub-tasks $q_1 \rightarrow a_1$ and $q_2 \rightarrow a_2$, the combined task $q_{width}$ can be represented as

$$(q_{width} = q_1 + q_2) \rightarrow a_1 + a_2, \tag{4}$$

where the $+$ indicates using LLM to merge and rephrase two question strings.

**Trajectory generation.** Two strategies exist for generating execution trajectories in this task: (1) For simple tasks, such as atomic tasks, existing agents can directly infer and capture the trajectory, including tool selection, parameters, return results, and plans. (2) For complex tasks, such as depth-wise extension tasks, the sub-task trajectory is recorded while iteratively expanding and validating new atomic tasks.

### 3.3 TASK VERIFICATION

To ensure that the generated tasks demand agentic reasoning and that each expansion is effective, a verification is performed after every step. Within this workflow, task verification can be carried out in two phases:

**Atomic task verification**: An atomic task is defined as a simple agent task solvable via a single tool call. During verification, we relax this definition slightly: for each candidate task, we evaluate the task agent's output within a limited number of tool-use steps (e.g., three) and compare it with an infer-LLM separately. A judge-LLM verifies whether only the agent's output contains the golden answer, retaining only validated tasks. (see Appendix E for more details)

**Task extension verification**: This process is conducted purely through linguistic analysis without agent involvement. During depth-wise extension, we first employ a judge-LLM to validate: (1) whether the obtained $i_T^{n+1}$ and its relation $R^{n+1}$ constitute a proper superset of $i_T^n$ with logically sound relationships, and (2) whether the final input index $i_T^n$ in $q^n$ is appropriately replaced by $\hat{q}^{n+1}$ in the expanded task $q^{n+1}$. Furthermore, an infer-LLM derives the merged task, while the judge-LLM filters out tasks where the correct result is easily inferred, preventing information leakage that could render the task trivially solvable after merging. (see Appendix D for more details).

This framework ensures efficiency by applying agent reasoning only in atomic task verification at creation, while relying on LLM-based verification elsewhere for faster execution. It also enables complex task generation beyond agent capabilities, with reverse reasoning providing supervisory signals to enhance agent learning or reinforcement learning.

## 4 EXPERIMENTS

### 4.1 CORPUS CONSTRUCTION

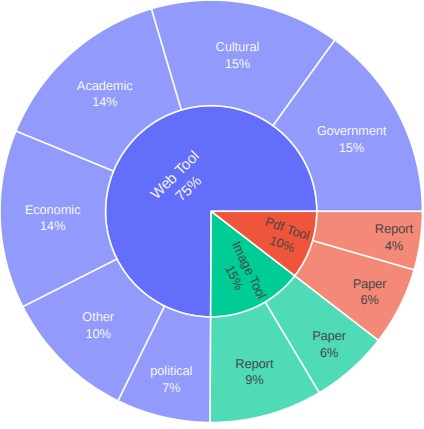

Figure 4: Corpus source distribution.

Table 1: Human evaluation for the generated tasks.

|  |  |  |
|---|---|---|
| Atomic | Linguistic fluency | 91.7% |
|  | Accuracy | 95.0% |
|  | Single answer | 83.3% |
|  | Information leakage | 11.7% |
| Depth-based extension | Extended validity | 82.3% |
|  | Non-superset | 8.5% |

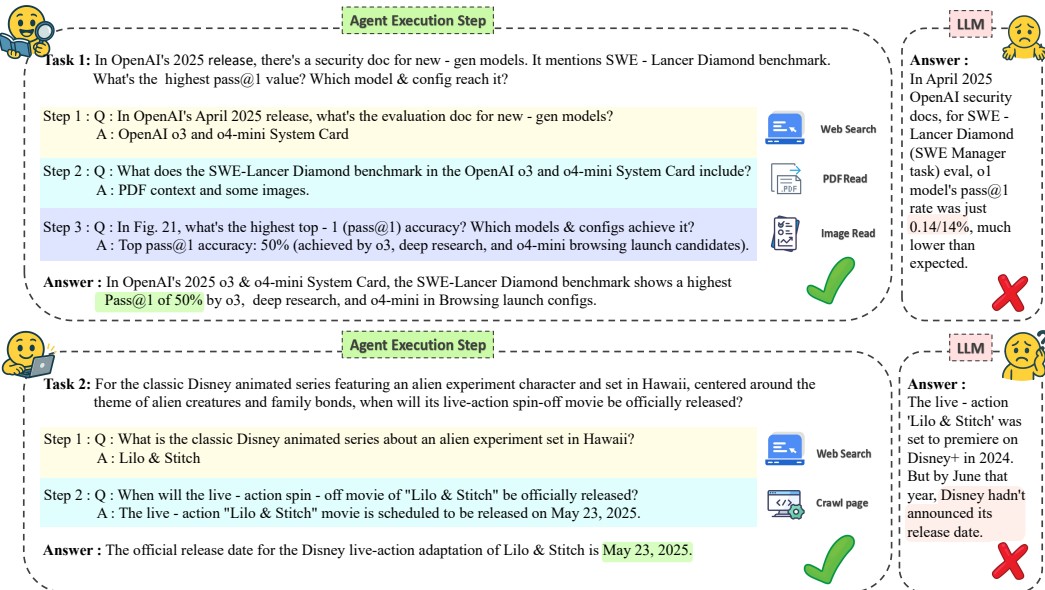

Figure 5: Generated case examples requiring multiple tool calls for completion.

We collect seed documents across modalities to generate tool-specific atomic tasks, extracting key insights for relevance. For instance, our PDF processor constructs atomic tasks by combining titles with core findings, enhancing the need for agent-based PDF tool invocation. To support atomic task generation, we constructed a dataset comprising webpages, PDF files, and images. Webpage data constitutes the largest proportion (75%), sourced from up-to-date news across multiple domains. Image data accounts for 15%, primarily derived from financial reports and research papers, with filtering to retain images containing information beyond text. PDF data makes up 10%, originating from English financial documents and academic publications.

**Human Evaluation.** To verify the validity of the results, we randomly sampled 60 atomic tasks and 48 depth-based extension tasks using human evaluation and scored them. As shown in Table 1, these results highlight the overall effectiveness and controllability of task generation.

## 4.2 ENHANCING TASK GENERATION EFFICIENCY VIA PROMPT LEARNING

As noted in section 3.3, both atomic task generation and task extension require validation, with any failure leading to rejection. Reducing rejections and improving efficiency requires refining four prompt designs that substantially affect the rejection rate:

- How to extract candidate answers from the corpus for atomic task generation (section 3.1).
- In depth-wise extension:
  - how the agent identifies the next input index $i_T^{n+1}$ relevant to the current document and avoid cyclic reference;
  - how to prompt the LLM to generate a relation $R^{n+1}$ so that the answer can be uniquely derived from the context;
  - How to integrate extended tasks and increase the complexity of existing questions, i.e., $(q^{n+1}, a) = (f_m(q^n, \hat{q}^{n+1}, i_T^n), a)$, while maintaining clarity and coherence.

We adopt bootstrap few-shot learning Khattab et al. (2024) to optimize the four prompts. For atomic task generation, the prompt is augmented with 20 randomly sampled examples. Multiple candidate prompts are evaluated, and the one yielding the highest pass rate is selected. For task extension, we focus on depth-wise augmentation and apply the same strategy using 10 sampled examples, refining the prompts to maximize the number of reasoning hops.

To enhance the LLM's capability in identifying intermediate objectives, we employ bootstrap few-shot learning Khattab et al. (2024) to systematically optimize four prompts corresponding to key challenges. Each prompt for atomic task generation is enhanced by appending 20 randomly sampled examples. Various prompt configurations are evaluated iteratively based on pass rates to select optimal examples. For depth-based extension, we optimize prompts using 10 randomly sampled examples, refining them to maximize task complexity.

Table 2: Effectiveness of generated task data in prompt learning and depth-wise extension. The pass rate denotes the proportion of atomic tasks that successfully pass validation out of all generated candidates. For depth-wise extension, the pass rate is defined as the fraction of successful extensions out of six attempts.

| Method | Pass rate | Time |
|---|---|---|
| Atomic Task | 54.9% | 29.1s |
| + Optimization | **68.1%** | **23.5s** |
| Depth-wise@6 | 41.0% | 31.5s |
| + Optimization | **51.2%** | **30.2s** |

Table 2 examines atomic task generation and depth-wise task extension before and after prompt learning, highlighting the role of generated tasks in enabling self-evolution within the workflow. These results validate the effectiveness of generated task data in enhancing sampling efficiency and supporting workflow adaptation. The optimized prompts are presented in Appendix E.2.

## 4.3 AGENT FOUNDATION MODEL TRAINING

| Method | SFT | RL | GAIA (%) | WebWalker | BrowserComp | HLE |
|---|---|---|---|---|---|---|
| **Qwen-2.5-7B-Instruct** | | | | | | |
| R1-Searcher Song et al. (2025) | ✓ | ✓ | 20.4 | - | - | - |
| WebSailor Li et al. (2025a) | ✓ | ✓ | 37.9 | - | 6.7 | - |
| 5k MHQA | ✓ | | 18.6 | 20.2 | 4.5 | 3.6 |
| 7.5k MHQA | ✓ | | 20.4 | 23.4 | 3.6 | 4.2 |
| 7.5k TaskCraft | ✓ | | 36.3 | **55.0** | 12.4 | **16.4** |
| 5k MHQA + 2.5k TaskCraft | ✓ | | 34.0 | 52.6 | 6.4 | 13.2 |
| **5k MHQA + 2.5k TaskCraft (SFT) + 8k TaskCraft (RL)** | ✓ | ✓ | **40.8** | - | **13.4** | 16.0 |
| **DeepSeek-R1-Distill-Llama-8B** | | | | | | |
| 7.5k MHQA | ✓ | | 21.6 | 28.6 | 3.6 | 9.6 |
| **5k MHQA + 2.5k TaskCraft** | ✓ | | **33.0** | **59.4** | **7.6** | **12.8** |
| **QwQ-32B** | | | | | | |
| Search-o1 Li et al. (2025b) | ✓ | ✓ | 39.8 | 34.1 | - | - |
| SimpleDeepSearcher Sun et al. (2025) | ✓ | ✓ | 50.5 | - | - | - |
| WebSailor Li et al. (2025a) | ✓ | ✓ | 50.5 | - | - | - |
| WebThinker Li et al. (2025c) | ✓ | ✓ | 48.5 | 46.5 | - | 15.8 |
| WebDancer Wu et al. (2025a) | ✓ | ✓ | 51.5 | 43.2 | 2.8 | - |
| **Qwen-2.5-32B-Instruct** | | | | | | |
| Search-o1 Li et al. (2025b) | ✓ | ✓ | 28.2 | - | - | - |
| SimpleDeepSearcher Sun et al. (2025) | ✓ | ✓ | 40.8 | - | - | - |
| WebSailor Li et al. (2025a) | ✓ | ✓ | 53.2 | - | 10.5 | - |
| 5k MHQA | ✓ | | 38.8 | 36.8 | 5.6 | 10.8 |
| 7.5k MHQA | ✓ | | 42.7 | 41.6 | 5.8 | 12.6 |
| 7.5k TaskCraft | ✓ | | 60.2 | - | 22.4 | 20.2 |
| 5k MHQA + 2.5k TaskCraft | ✓ | | 60.2 | - | 21.0 | 20.0 |
| **5k MHQA + 2.5k TaskCraft (SFT) + 8k TaskCraft (RL)** | ✓ | ✓ | **60.8** | - | **24.8** | **20.6** |

Table 3: Performance across agentic task benchmarks. Methods are grouped according to the base model adopted.

To validate the effectiveness of our synthetic tasks, we apply SFT and RL to refine a tool-integrated reasoning (TIR) model in agentic scenarios. In TIR, the LLM output is trained with explicit tags such as `<tool>`, `<observation>` or `<think>`, which structure the reasoning flow and trigger corresponding tool calls. We conduct experiments using models from different families and scales, evaluating their performance on the GAIA Mialon et al. (2023) , WebWalker Wu et al. (2025b), BrowserComp Wei et al. (2025), and HLE Phan et al. (2025).

For SFT learning, we sample solution trajectories for each task using Oagents Zhu et al. (2025), and convert them into the TIR model format. To ensure the performance gains are not merely due

to learning the output format, we use two types of training data: 7.5k tasks sampled from existing multi-hop QA datasets (denoted as MHQA, including HotpotQA and NQ), and 7.5k synthetic tasks via our pipeline. To further enhance model performance, we incorporate additional generated data and apply DAPO Yu et al. (2025) for continued RL training.

As shown in Table 3, pure 7.5k TaskCraft outperforms 7.5k MHQA across all benchmarks. Furthermore, replacing 2.5k MHQA with 2.5k TaskCraft produces 5–16× larger gains, far exceeding the improvements obtained by adding the same amount of MHQA. Even without RL, TaskCraft-trained models already match SOTA systems that rely on both SFT and RL. When scaled with more TaskCraft tasks and RL, performance further improves, reaching new SOTA. For example, on WebWalker, our Qwen-2.5-7B-Instruct exceeds the previous best—including the much larger QWQ-32B—by a substantial margin. These results confirm that TaskCraft is highly scalable and substantially enhances agent model performance, enabling models to reach SOTA levels.

## 4.4 EFFECTIVENESS OF TOOL CONTEXT IN CONSTRUCTING AGENTIC TASKS.

In atomic task generation, we incorporate the input index $i_T$ and the tool-answer relation $R$ to structure tasks. To evaluate its effectiveness, we conduct an ablation study where an LLM directly generates single-tool tasks $q$ without using $i_T$ or $R$. We assess performance via pass rate, resolution time, average tool usage, and usage variance.

Table 4: The effectiveness of tool context.

| Method | Pass rate | Time | #Tool-use | $\sigma^2$ |
|---|---|---|---|---|
| LLM only | 18.5% | 119.7s | 2.8 | 1.2 |
| **Ours** | **43.0%** | **86.7s** | **2.1** | **0.4** |

Compared to direct GPT-4.1 prompting, our method significantly improves atomic task generation, achieving higher success rates and faster task construction. It produces more atomic and consistent tasks, with fewer and more stable tool invocations, highlighting the limitations of vanilla LLMs in agentic task design and the robustness of our structured workflow.

## 4.5 SYNTHETIC TASKS ANALYSIS

**Agent reasoning analysis.** To practically assess task difficulty, we sample 1,000 tasks and deploy both Smolagents Roucher et al. (2025) and Oagents Zhu et al. (2025), for execution and validation. While both agents performed identical tasks, Oagents incorporated advanced tool capabilities for refined analysis.

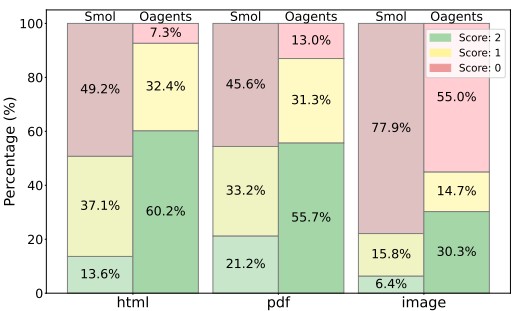

Figure 6: score distribution comparison

Responses were evaluated by comparing the agents' outputs to the golden answer, following a three-point scoring scheme: 2 for fully correct responses, 1 for answers that included the golden answer but contained additional information, and 0 for incorrect responses.

In figure 6, task failure rates increase from web pages to PDFs and then to images within PDFs, indicating that multi-hop web search tasks are more manageable for agents, while complex comprehension challenges, such as PDF extraction and image interpretation, remain difficult. Additionally,

these results demonstrate that our generated tasks span varying difficulty levels, including those that pose significant challenges for current agent capabilities.

**Scalability of TaskCraft data.** To evaluate TaskCraft's scalability, we trained Qwen2.5-7B-instruct models on 1k, 3k, and 5k randomly sampled tasks, using consistent settings and tested them on GAIA-103. As shown in Table 5, the results exhibit a clear upward trend, suggesting that larger TaskCraft training sets yield progressively better performance.

Table 5: GAIA performance by data size.

| Data Size | Pass@3 on GAIA-103 |
|-----------|---------------------|
| 1,000 | 17.5% |
| 3,000 | 31.1% |
| 5,000 | 39.8% |

Table 6: Accuracy comparison of Smolagents on GAIA and synthetic tasks.

| | Level1 | Level2 | Level3 | Avg. |
|---|---|---|---|---|
| GAIA | 54.71 | 43.02 | 26.92 | 44.20 |
| | PDF | html | Image | Avg. |
| Synthetic Task | 54.4 | 50.7 | 22.1 | 42.4 |

**Comparison with the GAIA Dataset.** Table 6 compares Smolagent's accuracy on the GAIA benchmark and our generated dataset. The results show that tasks derived from diverse tool corpora reflect GAIA's stratified difficulty levels, with image understanding tasks presenting the greatest challenge and yielding accuracy comparable to Level 3. Unlike GAIA, which relies heavily on manual annotation, our framework automates task generation—eliminating the need for labor-intensive labeling while preserving scalability and adaptability for agent self-evolution and optimization.

# 5 RELATED WORK

## 5.1 INSTRUCTION DATA GENERATION

Synthetic data has emerged as a promising solution for enhancing performance and enabling new capabilities. STaR Zelikman et al. (2024) augments learning with chain-of-thought (CoT) rationales but often requires a substantial number of task queries beforehand. Methods such as Self-Instruct Wang et al. (2022), Self-Chat Xu et al. (2023b), NuminaMath Li et al. (2024), and OpenMathInstruct-2 Toshniwal et al. (2024) generate data from minimal seed examples using LLMs, yet they struggle to extend task generation for multiple tool invocations. WizardLM Xu et al. (2023a) employs Evol-Instruct to incrementally enhance instruction complexity. However, it relies primarily on rule-based modifications, making its generated instructions unsuitable for agentic task scenarios. MetaMath Yu et al. (2023) generates mathematical data by rewriting questions, but adapting agent tasks to environmental feedback presents challenges beyond simple rephrasing. WebInstruct Yue et al. (2024) extracts question-answer pairs from a pre-training corpus across multiple domains; however, the generated questions often fail to incorporate tool utilization. AutoAct Qiao et al. (2024) uses a self-planning mechanism to generate planning trajectories for QA tasks.

## 5.2 LANGUAGE AGENT

Existing research on agentic task execution advances along two main axes: role specialization and functional partitioning. Role-based approaches, such as AutoGPT Significant-Gravitas (2023), AutoGen Wu et al. (2023), and Camel Li et al. (2023), organize collaborative agents by dynamically assigning tools. In contrast, frameworks like Barcelona2, Omne, and AgentIM[1] adopt functional partitioning to optimize modular efficiency. SmolAgents Roucher et al. (2025) integrates ReAct Yao et al. (2023) and CodeAct Wang et al. (2024b) into a hierarchical agent system for iterative code-based task execution. Magnetic-One Fourney et al. (2024) enhances multimodal performance by decoupling perception Yang et al. (2023a;b), planning Song et al. (2023); Tordesillas & How (2021), and execution Qin et al. (2024); Wang et al. (2024b) modules. Dynamic orchestration mechanisms address real-time adaptation and robustness. Trase-Agent Trase (2024) adapts strategies based on feedback, while TapeAgents Bahdanau et al. (2024) uses asynchronous communication to improve coordination. Studies show that stable sub-agent interactions outperform complex centralized orchestration. To advance autonomy, AutoAgent Tang et al. (2025) supports no-code agent customization via natural language coordination, modular workflows, and self-managing file systems. Hybrid

---

[1] These are closed-source frameworks.

systems like h2oGPTe-Agent H2O.ai (2024) explore multi-agent optimization, achieving strong results in code generation, though cross-modal bottlenecks remain a challenge.

## 6 CONCLUSION

We introduced TASKCRAFT, an automated workflow for scalable, multi-tool, and verifiable agentic task generation. By applying depth-based and width-based extensions to atomic tasks, the framework constructs hierarchically complex challenges with incremental validation. To enhance sampling efficiency, we incorporated a self-evolving prompt optimization strategy inspired by bootstrap few-shot learning. Experiments across multiple LLMs demonstrate that TaskCraft-generated data significantly improves multi-hop reasoning and agentic capabilities, achieving performance comparable to state-of-the-art RL models using only SFT. Further scaling and RL fine-tuning with TaskCraft tasks yield additional gains, culminating in state-of-the-art results on four agentic benchmarks. The final dataset comprises 41,000 tool-intensive tasks spanning diverse difficulty levels, including 12,600 tool-interaction trajectories and 5,000 multi-hop decompositions.

## 7 ETHICS STATEMENT

This research adheres to the ethical guidelines outlined by the ICLR conference. The proposed methods do not involve human subjects, personally identifiable information, or sensitive data. All datasets used are publicly available and widely adopted in the research community. No data was collected from vulnerable populations, and no deceptive practices were employed.

The model outputs were evaluated for fairness and robustness. We conducted thorough error analysis to ensure that the system does not propagate harmful biases or stereotypes. Where applicable, mitigation strategies were applied to reduce unintended consequences.

Our work does not pose foreseeable risks to individuals or society. We acknowledge that any deployment of agentic systems should be accompanied by safeguards to prevent misuse. We encourage future researchers and practitioners to consider the broader societal impact of autonomous agents and to adopt responsible AI practices.

## 8 REPRODUCIBILITY STATEMENT

To support faithful replication, we will release all artifacts referenced in this paper. Specifically: (*i*) the complete TASKCRAFT workflow code, encompassing atomic task generation, depth/width extensions, incremental validation, and rejection sampling, along with all associated prompts, templates, and data pre-/post-processing scripts; (*ii*) the full synthetic dataset comprising 41,000 tool-intensive tasks, formatted in a standardized JSON schema; and (*iii*) 12,600 tool-interaction trajectories and 5,000 multi-hop decompositions, to be released subsequently.

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

## A    USE OF LLMS

Large language models (LLMs) are used in this work exclusively for text polishing and language refinement during the writing process. Specifically, LLMs assist in improving the fluency, clarity, and conciseness of the writing.

LLMs are not used for any aspects of experimental design, methodological development or scientific interpretation. All scientific contributions and innovations presented in this work are entirely human-originated.

## B    DATA STATISTICS

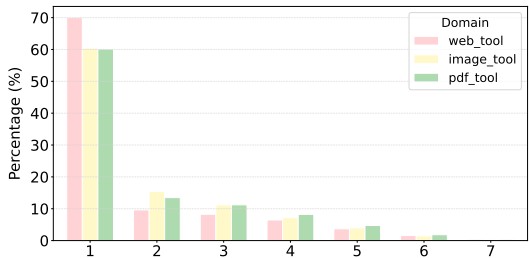

Figure 7: Analysis of all tasks.

As illustrated in figure 7, task generation exhibits a hierarchical decay pattern across all domains as hop count increases, revealing distinct scalability trends:

- **pdf_tool domain**: Shows gradual performance attenuation with hop depth, 1-hop tasks accounting for 60.13% (8,115 tasks), decreasing to 13.49% (1,820 tasks) for 2-hop and 11.22% (1,514 tasks) for 3-hop. The sharp drop in 5-7 hop tasks (6.94% combined) indicates limited deep-extension capability, yet surpasses other domains in depth scalability.

- **image_tool domain**: Presents the most pronounced performance decay, with 1-3 hops comprising 87.10% (7,125/8,180 tasks) but only 5.71% (467 tasks) for 5-7 hops, highlighting fundamental constraints in deep hierarchical task generation.

- **web_tool domain**: In the web_tool domain, 1-hop tasks dominate, constituting 70.01% (13,467 tasks) of the total. However, this domain also has the highest absolute number of deep extensions, with 5-7 hop tasks accounting for 5.66% (1,089 tasks).

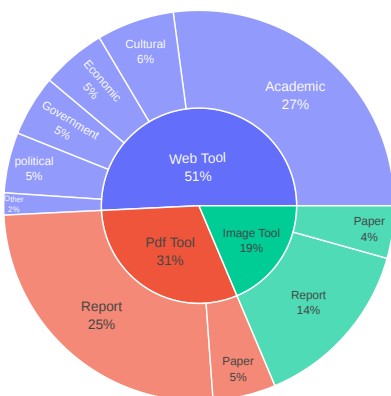

Figure 8: Distribution of atomic data.

**Atomic task analysis.** We collect data from webpages, PDF files, and images to support the generation of atomic tasks, which form the basis of the dataset, totaling 26,527 instances as shown in figure 8.

Among them, atomic conclusions from web-based tools account for the largest proportion, reaching 50.77%, with sources spanning multiple domains: academic (27.11%), cultural (6.42%), economic (5.36%) and governmental (5.05%) resources. These derive from up-to-date news and curated online materials for relevance.

Image-based tools contribute 18.64% of the data, extracting structured insights (e.g., key trends, comparisons) from charts/tables in financial reports and research papers. Strict verification excludes conclusions directly replicating source text to avoid redundancy.

PDF-based extraction accounts for 30.59%, supplementing the dataset with findings from financial reports and academic publications. This multi-source approach enhances diversity while maintaining consistency in atomic fact representation.

By systematically integrating these extraction methods, we ensure high-quality task generation, providing a robust foundation for downstream model training and optimization.

## C  EXPERIMENTS ON MULTI-HOP QA TASKS

We first evaluate our models across three established multi-hop question answering benchmarks: HotpotQA Yang et al. (2018), Musique Trivedi et al. (2023), and Bamboogle Press et al. (2023). These datasets present diverse challenges in reasoning and search, providing a robust evaluation platform.

We compare the baseline workflow (Search-R1 Jin et al. (2025b), which leverages reinforcement learning for LLM model optimization) with the agent workflow after applying SFT using the generated tasks.

| Method | HotpotQA | Musique | Bamboogle | Avg. |
|---|---|---|---|---|
| **Qwen2.5-3B-Base** | | | | |
| Search-R1 | 0.284 | 0.049 | 0.088 | 0.140 |
| **+ SFT** | **0.344** | **0.111** | **0.280** | **0.245** |
| **Qwen2.5-3B-Instruct** | | | | |
| Search-R1 | 0.324 | 0.103 | 0.264 | 0.230 |
| **+ SFT** | **0.340** | **0.104** | **0.264** | **0.236** |

Table 7: Performance across three datasets and two models. Avg. denotes average.

As shown in Table 7, our synthetic data proves valuable in SFT training, showing average performance improvements of +14.0% (Qwen2.5-3B-Base) and +6.0% (Qwen2.5-3B-Instruct) compared to their respective base workflows, validating our data generation approach. Compared to the Search-R1 baseline, the trained model demonstrates substantial improvements. This suggests that our synthetic data not only enhances immediate task execution but also optimizes RL initialization effectively.

## D  VERIFICATION REQUIREMENTS FOR DEPTH-BASED EXTENSION

Effective n-hop task extension requires rigorous verification to ensure valid multi-hop reasoning. The transformation must preserve superset validity:

$$(\hat{q}^{n+1}, i_T^n) = (f_q(i_T^{n+1}, R^{n+1}), i_T^n). \tag{5}$$

$$(q^{n+1}, a) = (f_m(q^n, \hat{q}^{n+1}, i_T^n), a), \tag{6}$$

Current depth-based extension methods often introduce two critical flaws when replacing tool inputs $i_T$ without proper verification:

- **Pseudo-Superset Task**: Superficial substitutions that preserve semantic equivalence but lack genuine superset relationships

- **Information Leakage**: Premature disclosure of information that should only emerge through proper multi-step reasoning

These issues undermine the intended multi-hop reasoning process.

## D.1 PSEUDO-SUPERSET TASK

A fundamental limitation arises when replacing $i_T$ with a semantically equivalent but non-superset index $i_T^{n+1}$. Consider the following task extension example:

> **Original task**
>
> **Input index ($i_T$):** Travel Trends 2025 — Our Annual Report
> **Query ($q^n$):** How many travel trends for 2022 does 'Travel Trends 2025 — Our Annual Report' present?
> **Answer:** 5

When the search agent retrieves the superset $i_T^{n+1}$ of $i_T$, it actually ends up retrieving the synonyms of $i_T$ instead. Based on this, an intermediate task is derived:

> **Intermediate task**
>
> **New input index ($i_T^{n+1}$):** 2025 Annual Travel Trends Report
> **Query ($\hat{q}^{n+1}$):** What is the title of 2025 Annual Travel Trends Report?
> **Answer :** Travel Trends 2025

Despite valid hop annotations, the intermediate question does not constitute an effective extension: it does not represent a necessary tool-use step. The core issue lies in the absence of a genuine superset relationship between $i_T^n$ and $i_T^{n+1}$, leading to superficial expansion.

> **Extended task**
>
> **Query ($q^{n+1}$):** How many travel trends for 2022 does '2025 Annual Travel Trends Report' present?
> **Answer:** 5

## D.2 INFORMATION LEAKAGE

A second failure mode occurs when expanded tasks inadvertently expose original answers, enabling large language models (LLMs) to bypass tool retrieval. For instance, consider the extended task:

> **Original task**
>
> **Input index ($i_T$):** Sports In Brief
> **Query ($q^n$):** What is the merger value of Charter and Cox in the Sports In Brief?
> **Answer:** 34.5B USD

> **Intermediate task**
>
> **New input index ($i_T^{n+1}$):** AP News
> **Query ($\hat{q}^{n+1}$):** What is the section in AP News that updates sports news daily?
> **Answer :** Sports In Brief

> **Extended task**
>
> **Query ($q^{n+1}$):** In the AP Sports daily summary, Charter and Cox's proposed merger is valued at approximately \$34.5 billion. What is the exact amount?
> **Answer :** 34.5B USD

While $q^{n+1}$ appropriately conceals the previous $i_T^n$ ("Sports In Brief"), it directly reveals the answer "34.5B USD", allowing the LLM to bypass the intended retrieval process. This compromises the essential tool dependency required for multi-hop task answering.

### D.3  VERIFICATION FOR TASK EXTENSION

To address these challenges, we propose a rigorous verification framework to ensure the validity of $i_T^{n+1}$, $\hat{q}^{n+1}$ and $q^{n+1}$ in task extension.

#### D.3.1  STRICT SUPERSET VERIFICATION

$i_T^{n+1}$ must be the index of a strict superset of $i_T^n$, and the relationship can be formalized as:

$$(\hat{q}^{n+1}, i_T^n) = (f_q(i_T^{n+1}, R^{n+1}), i_T^n). \tag{7}$$

where $R^{n+1}$ denotes hierarchical relations (e.g., *contains*, *part_of*). Valid extensions must introduce genuine depth, such as *"Sports In Brief"* $\rightarrow$ *"AP News"* (relation: *the part that updates sports news daily*), while rejecting synonymous substitutions. Additionally, invalid extensions that allow the LLM to derive $i_T^n$ directly should be excluded—ensuring the intermediate task $\hat{q}^{n+1}$ requires tool use.

#### D.3.2  INFORMATION LEAKAGE VERIFICATION

$$(q^{n+1}, a) = (f_m(q^n, \hat{q}^{n+1}, i_T^n), a), \tag{8}$$

The extended query $q^{n+1}$ must adhere to the information-sealing principle to ensure proper tool-use reasoning. This requires that the query does not directly expose the original answer, and any query from which the LLM can directly obtain the answer should be filtered out.

### D.4  ADVANTAGES OF THE VERIFICATION FRAMEWORK

Our approach provides three key advantages:

- **Superset Integrity**: Guarantees valid hierarchical progression (e.g., *column* $\rightarrow$ *page* $\rightarrow$ *website*) without logical gaps.
- **Strict Tool Dependency**: Enforces authentic multi-hop reasoning by eliminating solution shortcuts, ensuring mandatory tool-use.
- **Transparent Reasoning**: Offers full explainability through explicit relation paths ($R^n$).

Below shows the intermediate extension process from 1-hop tasks to 4-hop tasks, so as to highlight the increasing difficulty of the tasks:

---

**Original Task (n=1)**

**Input index** ($i_T^1$): 2024's Rising Content and Fastest Growing Skills for 2025
**Query** ($q^1$): What percentage of non-job seekers see the value of AI upskilling according to '2024's Rising Content and Fastest Growing Skills for 2025'?
**Answer :** 49%

---

**Extended Task (n=2)**

**New input index** ($i_T^2$): Coursera Blog
**Intermediate Query** ($\hat{q}^2$): Which Coursera Blog article covers 2024 content trends and 2025 growing skills, and is easily identifiable on the blog's homepage?
**Query** ($q^2$): Referring to the article on the Coursera Blog that discusses 2024 content trends and 2025 growing skills, and can be uniquely identified from the blog homepage, what percentage of non-job seekers recognize the value of AI upskilling according to its findings?
**Answer :** 49%

---

---

**Extended Task (n=3)**

**New input index** $(i_T^3)$: Coursera
**Intermediate Query** $(\hat{q}^3)$: What is the official information and content update section of the Coursera online learning platform, which is a content subset and can be accessed as a dedicated section on the main Coursera website?
**Query** $(q^3)$: Referring to the official information and content update section of the Coursera online learning platform, which is a content subset available as a dedicated section on the main Coursera website and features discussion of the 2024 content trends and 2025 growing skills, and can be uniquely identified from the platform's homepage, what percentage of non-job seekers recognize the value of AI upskilling according to its findings?
**Answer :** 49%

---

**Extended Task (n=4)**

**New input index** $(i_T^4)$: Global online learning platform
**Intermediate Query** $(\hat{q}^4)$: Which global online learning platform collaborates with over 275 leading universities and companies to offer MOOCs and degree programs, enabling users to access and identify publicly available course content from authoritative educational institutions in one place?
**Query** $(q^4)$: Referring to the official information and content update section of the global online learning platform that collaborates with over 275 leading universities and companies to provide MOOCs and degree programs, and which offers a dedicated content subset as a section easily identifiable on its homepage—including discussion of the 2024 content trends and 2025 growing skills—what percentage of non-job seekers recognize the value of AI upskilling according to the findings available there?
**Answer :** 49%

---

# E    CORE PROMPTS

This section presents key prompts used in our framework.

## E.1    ATOMIC TASK VERIFICATION

The following prompt is used in atomic task verification (Section 3.3):

---

**Atomic task verification**

**Task**: Evaluate the *consistency* between the golden answer (GA) and another answer (AA, either agent or LLM-generated) as follows:

- **2 points (Fully Consistent)**: AA and GA are semantically equivalent, even if phrased differently.
  *....(Example)....*

- **1 point (Partially Consistent)**: AA includes all GA information but adds valid extra details.
  *....(Example)....*

- **0 points (Inconsistent)**: AA omits key GA information or contradicts it.
  *....(Example)....*

The criteria prioritize semantic equivalence while accommodating informative expansions or reductions.
......

---

A task is retained as an atomic task if and only if the *AgentScore* strictly exceeds the *LLMScore*.

## E.2    OPTIMIZED PROMPTS

The following prompts is optimized prompt mentioned in (Section 4.2):

---

**Atomic Conclusion Extraction**

**Task**: Extract standalone conclusions from document chunks meeting these criteria:

1. **Atomicity**: Extract only indivisible basic facts ....*(Example)*....

2. **Verifiability**: Include at least one definite identifier (numeric value, time, unique name) and reject vague expressions ....*(Example)*....

3. **Timeliness Handling**: Explicitly mark time ranges for time-sensitive information ....*(Example)*....

4. **Citation Integrity**: Embed complete content of cited references ....*(Example)*....

....*(Example)*....

---

**Depth-wise Extension with $i_T^{n+1}$ and $R^{n+1}$**

**Task**: Identify a minimal unique superset for an input element based on its attributes, ensuring the superset+relationship uniquely points to the element.
....*(Example)*....
**Relationship expression guidelines**:

1. Clearly show hierarchical/ownership. Indicate position for series sub-items; clarify ownership for parts of a superset

2. Specify input content's positioning (e.g., time range, publication field, role in superset)

3. Use research/industry standard wording

4. Provide only necessary associations

....*(Example)*....

---

**Logical Substitution: $(q^{n+1}, a)$ as $(f_m(q^n, \hat{q}^{n+1}, R^n), a)$**

**Task**: Substitute elements in core queries using auxiliary queries while preserving:

1. **Complexity Balance**: The new query should be slightly more complex than the original core Query and require more steps to solve. But do not make too many changes to the core query.

2. **Answer Uniqueness**: The new query should point to the unique answer: golden answer, and should not point to other answers.

3. **Answer Concealment**: The new query must not reveal information about the golden answer.

4. **Natural Language Polish**: After merging, polish the question to make it conform to human expression habits without changing the original meaning. Do not modify the proper nouns appearing in it.

....*(Example)*....

---

### E.3 STRICT SUPERSET VERIFICATION

The following prompt is used in Appendix D.3.1:

> **Strict Superset Verification**
>
> **Task**: Verify if index $i_T^{n+1}$ uniquely determines subset $i_T^n$ under relation $R^n$ in given queries.
> **Criteria**:
>
> 1. **SupersetSubset Relationship**:
>    - $i_T^{n+1}$ must be the index of a superset that properly contains $i_T^n$
>    - $i_T^{n+1} \not\approx i_T^n$ (excluding synonym pairs like CAR/AUTOMOBILE)
> 2. **Relationship Validity**:
>    - The relationship $R^n$ must explicitly and uniquely link the superset to the subset (no many-to-one mappings)
>
> ......

## F  TOOL DETAILS

Our main tools are implemented as follows:

**Wiki Search Tool**: We use the same local WiKi Search tool as Search-R1 Jin et al. (2025a), which uses the 2018 Wikipedia dump Kaelbling et al. (1996) as the data source and E5 Wang et al. (2024a) as the retriever.

**Web Search Tool**: We employ a mechanism to access the Google search engine for information retrieval. Specifically, Serpapi (`https://google.serper.dev/search`) is utilized to execute web search operations. The core parameters configured for Serpapi include the search query string and the specified number of results to be returned. In practice, searches are conducted using queries generated by the model, with the system set to retrieve the top 10 results for each query. Each result contains a title, a snippet, and the corresponding URL. This setup furnishes substantial support for subsequent analytical processes and decision-making actions.

**Web Page Crawling Tool**: We implement a web page crawling tool with content summarization capabilities. The tool accepts three core parameters: target URLs, web search queries, and reasoning context. Each URL is processed using Jina (`https://jina.ai/`) to extract information. We then use the Qwen2.5-72B-instruct model to generate summaries for each crawled page. These summaries are concatenated based on the reasoning context to form the tool's output. Importantly, our summary prompt instructs the model to preserve relevant URLs, enabling iterative use of the crawling tool for deeper web exploration.

**PDF Tool**: We use pdfplumber with multiprocessing to read text from PDFs in parallel, and PyMuPDF to extract images from PDFs.

## G  FURTHER TRAINING DETAIL

For SFT training, we synthesize 3,202 multi-hop tasks and their trajectories and apply content masking to search tool contexts in these trajectories.

For RL training, we follow the Search-R1 Jin et al. (2025b) and use the 2018 Wikipedia dump as a knowledge source and the E5 embedding model as a retriever. For fair evaluation, we fix the retrieval depth to 3 passages for all methods. We merge the training sets of NQ and HotpotQA to form a unified dataset. Evaluation is conducted on the test or validation sets of three datasets to assess both in-domain and out-of-domain performance. Exact Match is used as the evaluation metric. In the PPO settings, we set the learning rate of the policy LLM to 1e-6 and that of the value LLM to 1e-5. Training is conducted for 500 steps, with warm-up ratios of 0.285 and 0.015 for the policy and value models, respectively. We use Generalized Advantage Estimation with parameters $\lambda = 1$ and $\gamma = 1$. We employ vLLM for efficient LLM rollouts, configured with a tensor parallelism degree of 1 and a GPU memory allocation ratio of 0.6. Our sampling strategy utilizes a temperature parameter of 1.0 and top-p threshold of 1.0. For policy optimization, we apply KL divergence regularization with coefficient $\pi=0.001$ and implement a clip ratio $\epsilon=0.2$. The action budget is constrained to 4, with a default retrieval depth of 3 passages per query.

